# A Five-Year Update on Matrix Compounds for MALDI-MS Analysis of Lipids

**DOI:** 10.3390/biom13030546

**Published:** 2023-03-16

**Authors:** Jenny Leopold, Patricia Prabutzki, Kathrin M. Engel, Jürgen Schiller

**Affiliations:** Faculty of Medicine, Institute for Medical Physics and Biophysics, Leipzig University, Härtelstr. 16/18, D-04107 Leipzig, Germany

**Keywords:** MALDI-MS, matrix, lipid, phospholipid

## Abstract

Matrix-assisted laser desorption and ionization (MALDI) is a widely used soft-ionization technique of modern mass spectrometry (MS). MALDI enables the analysis of nearly all chemical compounds—including polar and apolar (phospho)lipids—with a minimum extent of fragmentation. MALDI has some particular advantages (such as the possibility to acquire spatially-resolved spectra) and is competitive with the simultaneously developed ESI (electrospray ionization) MS. Although there are still some methodological aspects that need to be elucidated in more detail, it is obvious that the careful selection of an appropriate matrix plays the most important role in (lipid) analysis. Some lipid classes can be detected exclusively if the optimum matrix is used, and the matrix determines the sensitivity by which a particular lipid is detected within a mixture. Since the matrix is, thus, crucial for optimum results, we provide here an update on the progress in the field since our original review in this journal in 2018. Thus, only the development during the last five years is considered, and lipids are sorted according to increasing complexity, starting with free fatty acids and ending with cardiolipins and phosphoinositides.

## 1. Introduction

In a similar way as electrospray ionization (ESI) mass spectrometry (MS), matrix-assisted laser desorption/ionization (MALDI) time-of-flight (TOF) MS is another MS method that is widely used in medicine and biology due to its high sensitivity and the simple performance. As we have comprehensively outlined in our originally published manuscript [1], the matrix has a considerable effect on the sensitivity by which certain analytes, such as lipids, are detectable. Some compounds are only detectable if the most suitable matrix is used. This short amendment to our paper [1] gives a survey about the progress during the last five years (beginning with 2018) regarding the analysis of lipids and particularly lipid mixtures as they occur in body fluids such as blood and biological tissues, for instance, adipose tissue. The interest in understanding the physiological role of lipids is continuously increasing and was recently reviewed [2].

Further details about analytical approaches in the lipidomics field are available in, e.g., the recently published review by Han and Gross [3]. Papers with a focus on MALDI-MS are also available and are continuously updated [4,5].

### MALDI MS and the Role of the Matrix

The MALDI process works only in the presence of a matrix, which absorbs the emitted energy by the laser (in the majority of cases, an ultraviolet (UV) laser) and leads, thus, to the generation of sufficiently stable ions from the analyte. There are three different classes of matrices available:Typical organic matrices derived from cinnamic or benzoic acid. These compounds possess both an aromatic ring to absorb the UV laser light and acidic functional groups to enable the ionization of the analyte(s).Liquid crystalline matrices (for instance, norharmane combined with α-cyano-4-hydroxycinnamic acid (CHCA) [6]), which are useful if particular soft ionization is needed (e.g., for sulfated lipids which tend to undergo sulfate loss). However, the interest in this matrix class has been decreasing for a few years because applications are limited.Inorganic matrices such as graphene (or suitable derivatives such as graphene oxide [7]) or metal particles that provide only a weak background [8]. This is helpful for small molecules, including free fatty acids (FFA) that easily interfere with typical matrix peaks. One alumina-based material called DIUTHAME became recently commercially available and was shown to offer an improved reproducibility in comparison to 2,5-dihydroxybenzoic acid (DHB) [9], one of the most widely used MALDI matrices.

Our focus will be on (i) organic matrices that are still used in the majority of cases and (ii) UV lasers with emission wavelengths (λ) of either 337 or 355 nm. The energy of UV light with λ = 355 nm is reduced compared to 337 nm. This supports the MS analysis of compounds with functional groups that are easily lost (such as sulfate residues [10]). Further details about the criteria of laser selection can be found in the detailed book by Franz Hillenkamp (one of the inventors of the MALDI method) and Jasna Peter-Katalinić [11].

## 2. The Role of the Matrix

Despite the huge number of studies dedicated to (i) improved understanding of the role of the matrix and (ii) finding the most suitable matrix for a dedicated analytical problem, the finding of the optimum matrix is normally still an empirical process [12]. The optimum matrix should result in (i) a reasonable signal-to-noise (*S*/*N*) ratio, (ii) minimum analyte fragmentation, and (iii) moderate background signals to minimize interferences between the matrix background and peaks of interest. The last aspect is particularly challenging for small molecules [13], including lipids. The improvement of the homogeneity of the co-crystals between the analyte and the matrix is another important issue affecting the reproducibility of the spectra and, thus, of paramount importance for quantitative data analysis. There are many reviews [14,15] dedicated to this topic. Thus, this aspect will be scarcely discussed here.

### Commonly Used MALDI Matrices

Many chemicals were suggested in the past as potential MALDI matrices. However, only a very few of them became established, widely used matrix compounds [16]. We will focus here on lipid analysis and discuss the most important matrices in this field. The structures of the matrices that will be mentioned here are summarized in Figure 1.

## 3. Which Matrix Is Most Suitable for Individual Lipid Classes?

MALDI is typically used in analyzing polar molecules (such as DNA, peptides and, most typically, proteins), and there is only a handful of papers dealing with the MALDI analysis of lipids [4]. As in our original paper [1], we will discuss the most relevant lipid classes according to their complexity, i.e., we will start with FFA and end with complex glycerolipids and glycerophospholipids (called phospholipids (PL)). As already mentioned, the focus is on the developments since 2018.

There is a continuously increasing interest in MALDI-MS imaging (MSI) studies, which has some particular requests on the matrix. For instance, the vacuum stability of the matrix is even more important since MALDI MSI experiments take much more time than the acquisition of a conventional MALDI spectrum. For further details, please check the recently published reviews (for instance, [4,12,17]) or some textbooks with a focus on MSI (e.g., [18]).

### 3.1. Free Fatty Acids

MALDI-MS analysis of FFA is rather challenging if common organic MALDI matrices (such as DHB or CHCA) are used. One particular problem is the inevitable presence of massive background signals stemming from matrix-derived peaks, which show marked interferences with the signals generated by the FFA of interest. To give the reader an impression of this problem, some of the (many) DHB-derived peaks are assigned in [19].

One promising new matrix for the MALDI-MS analysis of FFA is 1,6-diphenyl-1,3,5-hexatriene (DPH) [20]. If low laser fluences are used, background signals of DPH are negligible. DPH is not only useful for the investigation of FFA but also provides structural characterization in more complex glycerolipids and PL. Using higher laser fluences, cleavage of the ester linkages in lipid ions occurs, leading to information about the released fatty acids. In addition to FFA, the majority of acidic PL (such as lysophospholipids, phosphatidic acid (PA), phosphatidylethanolamine (PE), phosphatidylserine (PS), phosphatidylglycerol (PG), phosphatidylinositol (PI), and sulfatides (ST)) are detectable in the presence of DPH matrix. In a very recent paper, the application of cyanographene [21] was suggested to monitor fatty acids (either as FFA or as bound in glycerolipids) in artwork paintings. Sensitivities of lower than 100 ng of lipids could be achieved.

Isomers of FFA are of increasing interest, and it was suggested that some of them are indicative of diseases [22]. Thus, the determination of the double-bond position(s) is of significant interest. Zhang and colleagues used peracetic acid [23] to generate epoxides from unsaturated fatty acids. The MS analysis of these epoxides gave abundant fragment ions, which can be used to determine the positions of the double bonds. Other approaches make use of ozonolysis [22,24] or the Paternò-Büchi reaction [25]. CeO_2_ and TiO_2_ nanopowders, in combination with ozonolysis, are also known to enable the scission of double bonds under conditions of UV irradiation [26]. Some oxidation products of FFA (such as chain-shortened aldehydes, for instance, nonanal) can hardly be detected by MALDI-MS because they are lost in the vacuum. “Reactive matrices”, which convert the volatile analyte into more readily detectable ions, can help to overcome this problem. This interesting topic has been recently reviewed [27].

### 3.2. Cholesterol and Cholesteryl Esters

Due to the comparably small molecular weight of cholesterol and its esters, the overlap with the matrix background—as already discussed in the FFA paragraph (vide supra)—is one major problem. Another problem is (as often observed in the case of apolar analytes) the poor ion yield of cholesterol. Thus, methods to overcome this problem were recently introduced [28]: first, cholesterol was converted into 4-cholesten-3-one using the enzyme cholesterol oxidase. The newly generated carbonyl group reacts with 4-hydrazinoquinazoline as a reactive matrix to form a hydrazone. It could be demonstrated that the ionization efficiency of cholesterol is considerably improved by this approach. A comparison of six different typical organic MALDI matrices had also been recently performed to elucidate the optimum matrix for cholesterol and phospholipid analysis by MALDI and MSI [29].

Although the method does not make use of a conventional MALDI matrix, it was recently demonstrated that surface-assisted laser desorption/ionization (SALDI) is an effective branch of MALDI. SALDI has been successfully applied to improve the detection of small molecules (including but not limited to lipids). The most important advantage of SALDI is the missing interferences with the matrix background. The authors [30] used a slightly modified polyvinylidene fluoride (PVDF) layer and obtained convincing results — particularly in the context of apolar lipids. Finally, the application of silver oxide-based nanoparticle-assisted laser desorption/ionization (AgLDI) MSI enabled the quantitative analysis of cholesterol from thin brain tissue sections at which cholesterol-d_7_ served as an internal standard [31]. A comprehensive survey of metal-assisted laser desorption/ionization MSI is available in [32].

### 3.3. Glycerolipids and Glycerophospholipids

Many lipids and PL merit considerable clinical and diagnostic interest [33,34]. Here, we will focus on the MALDI-based analysis of glycerolipids and PL.

#### 3.3.1. Di- and Triacylglycerols

Triacylglycerols (TAG) are important constituents of food (e.g., vegetable oils) and the biological storage form of excessive energy (adipose tissues), while diacylglycerols (DAG; normally generated from PL under the catalysis of the enzyme phospholipase C [35]) are relevant as lipid second messengers [36]. Suppression of positively charged DAG ions by other more abundant and readily ionizable lipid classes such as phosphatidylcholine (PC) is a particular problem in MALDI MSI studies since no separation (for instance, by different chromatographic approaches like high-performance liquid chromatography (HPLC)) of the individual lipid classes is possible prior to MS analysis. This problem was recently addressed [37], and it was shown that N-alkylpyridinium quaternization (in the presence of pyridine) of hydroxyl groups increases the ion yields of poorly detectable lipids. This is particularly helpful for monitoring minor constituents of biological mixtures such as DAG.

MALDI-TOF MS is excellently suitable for the fast screening (fingerprinting) of the compositions of crude TAG mixtures, for instance, vegetable oils [38] or food in general. The method is also suitable for monitoring the potential adulteration of expensive vegetable oils (such as extra virgin olive oil) by less expensive oils, which can be monitored by the changed fatty acyl composition [39]. The discrimination between bovine milk and non-dairy milk by MALDI lipid fingerprinting is another important example [40]. Little progress regarding the most suitable matrix has been made since the first TAG analysis by MALDI. DHB and super-DHB (a 9:1 mixture of DHB and 2-hydroxy-5-methoxybenzoic acid) were used in the majority of cases [41]. The poor interest in matrix optimization is presumably due to the fact that the majority of TAG is available in major amounts, and there is no need to increase the sensitivity of detection.

There are also forensic studies available [42] where the age of a fingerprint (in particular, the lipids within a fingerprint) can be determined by the O_3_-induced oxidation of TAG subsequent to ozonolysis [43]. The considerable relevance of MS-based lipidomics, including MALDI-TOF MS approaches in the meat industry, has been recently emphasized and comprehensively reviewed [44]. Recently, there have been strong efforts to obtain energy and fuel from algae (“renewable primary products”). A typical bioenergy material obtained from algae is the lipids family—often with polyunsaturated fatty acids that also have considerable nutritional value. The lipid compositions of algae can be screened by mass spectrometry resulting in characteristic fingerprints [45,46]. A survey of the different lipids in algae is available in [47].

#### 3.3.2. Phospholipids

PL are (in addition to cholesterol) the main membrane constituents of cells and tissues. Although many analytical methods are capable of resolving the structures of lipids, the lipid composition is most efficiently elucidated by MS—often but not necessarily in combination with chromatography [3,48]. The number of MALDI-TOF MS-based lipid studies is continuously increasing [4]—among other factors surely due to promising MSI approaches. MSI is a powerful technique to determine the spatially-resolved lipid composition of biological tissues [49]. Cellular membranes consist of considerable amounts of PL and are, thus, readily detectable by MSI experiments. In comparison, proteins are much more difficult to detect due to both their lower concentration and reduced ion yields according to their higher molecular weight. Since lipids are abundant constituents of tissues and can be easily detected by MS, lipids are increasingly coming into research focus. This is remarkable because the importance of lipids was neglected over decades. Another advantage of lipids is that these are species-independent compounds that occur in virtually all vertebrates (e.g., in mice and humans).

However, the demand for a universally usable PL matrix has not been satisfied yet. The choice of the most suitable matrix clearly depends (i) on the details of the scientific question, (ii) the number of the lipid species of interest, and (iii) the headgroups of the relevant lipids [50]. Especially point (iii) is important for the independent application of positive and negative ion detection.

Apart from a few exceptions, the majority of lipids are detectable mostly independent of the applied matrix. However, the different sensitivities by which the individual PL are detectable complicate the analysis of the positive and negative ion MALDI mass spectra. Thus, suitable internal standards (one standard per lipid class) are mandatory to obtain absolute quantitative data [51]. If no suitable internal standard is available or cannot be applied to the sample, the comparison of the relative intensities of selected peaks may be helpful. However, this has the disadvantage that it is not clear whether the intensities of both peaks or just a single one is changing [52]. Low-resolution (LR) shotgun (without previous separation) MALDI-MS is often considered to be insufficient to screen the compositions of complex mixtures. However, a comprehensive study [53] has recently shown that “…the simple shotgun LR-MS setup may also be considered for future screening because this configuration is well established in newborn screening.” 9-aminoacridine (9-AA, in a methanol-water mixture (4:1, *v*/*v*)) was used as a matrix in a concentration of 5 mg/mL. The extraction method of the lipids from the sample was also recognized to be very important [54].

There are many papers that emphasize the general importance of the matrix and indicate that particular analytical problems require the optimum matrix. The impact of the matrix on the shape of the mass spectra has been studied only recently [55]. Five different matrices (9-AA, 5-chloro-2-mercaptobenzothiazole (CMBT), 1,5-diaminonaphthalene (DAN), 2,5-dihydroxyacetophenone (DHAP), and DHB) were compared. Although positive ion detection was also affected by the used matrix, the most significant differences were found in the negative ion mode. Although a more detailed discussion of these aspects is beyond the scope of this review, the idea that all lipids are detected to the same extent in a single spectrum is surely not valid. However, impurities arising from the sample preparation (for instance, plasticizers from the used plastic vessels) can be completely masked by certain matrices such as 9-AA since this matrix detects exclusively charged (i.e., zwitterionic) compounds [4].

Recently, Wang and Hsu provided a comprehensive paper dealing with the evaluation of in-source fragmentation processes during the MALDI process [56]. In this study, the structural identification of PL and sphingolipids was achieved by the selective cleavage of the fatty acyl residues at the double-bond position without any additional (oxidative) modification but only with DHB and 9-AA as matrices. This method does not necessitate a dedicated collision cell but can be used at all mass spectrometers, even older devices. Selected spectra to illustrate the power of this method are shown in Figure 2:

### 3.4. Selected Examples of Typical Analytical Problems

#### 3.4.1. Oxidized Lipids

Some lipid oxidation products are only detectable by MALDI-MS if the optimum matrix is used. This is a very important aspect because the epilipidomics field is currently attracting considerable interest since oxidized lipids are involved in a number of (inflammatory) diseases [57]. The majority of studies of oxidized lipids apply ESI MS [58] because the ESI ionization process is softer compared to MALDI and, thus, the detection of labile compounds (such as hydroperoxides which tend to the loss of water) is improved [59].

The detection of oxidized lipids is a weak point of MALDI-TOF MS. This particularly applies to MSI, where studies on lipid oxidation products are—to these authors’ best knowledge—so far completely missing. Derivatization (although somewhat old-fashioned) is often helpful for the detection of oxidized compounds: for instance, 3-monochloropropane-1,2-diol (3-MCPD) is a well-known contaminant generated during thermal food processing. Unfortunately, 3-MCPD is rather difficult to detect by MS and particularly MALDI-MS. This study [60] has shown that derivatization of 3-MCPD with a boronic acid-modified C60 (B-C60) through the boronic acid-diol reaction facilitates the detection of 3-MCPD significantly. A detection limit of 9 ng could be achieved under these experimental conditions. The determination of the extent of lipid oxidation is commercially particularly important regarding the evaluation of the quality of vegetable oils. Fatty acids and chain-shortened fatty acids are typical products of unwanted lipid oxidation. Unfortunately, such products are only poorly detectable with conventional MALDI matrices. Dutta and coworkers [21] were able to show that cyanographene is a very powerful matrix in this field. A very comprehensive study in which PC oxidation products are detectable by MS has been recently published [61]. Although the interest in oxidized lipids is increasing, little progress has been made regarding improved MALDI matrices. DHB and 9-AA are most widely used for positive and negative ion detection, respectively [62]. Since the concentration of oxidized lipids is rather poor in biological systems, derivatization is even nowadays a convenient method to improve the detection of oxidized lipids such as chain-shortened aldehydes [27].

#### 3.4.2. Phosphorylated Phosphatidylinositols

Little progress has been achieved since our former review and, thus, during the last few years. HPLC-MS is still the most suitable method to detect the different phosphorylated PI (PIP). Since PIP faces the problem of poor ion yields in both MALDI-MS as well as ESI MS, derivatization (for instance, with CH_2_N_2_) is even nowadays commonly used to increase the ion yield [63].

#### 3.4.3. Cardiolipins

Cardiolipins (CL) possess a higher molecular weight than common PL and have two (negatively charged) phosphate groups. These peculiarities require improved matrices. Yang and coworkers introduced norharmane as an excellent matrix to detect CL. Since MSI was the focus of these studies, a sensitivity of 4.7 pg/mm^2^ was calculated [64]. The authors stated that neither 9-AA nor 1,8-bis(dimethylamino)naphthalene (DMAN) nor DHB allowed the detection of CL at these conditions. There are also indications that the use of non-conventional (non-organic, for instance, nanomaterials) matrices is beneficial in MSI studies of lipid distributions within tissue slices and enables improved detection [13]. This aspect was already reviewed [65]. Anyway, the MALDI-based search for biomarkers is massively ongoing: bis(monoacylglycero)phosphate, a peculiar PL with structural similarity to CL, could be recently introduced as a specific lipid marker of exosomes [66].

### 3.5. Glycolipids

Many different matrices, including CHCA and DAN, were already used to detect complex glycolipids such as gangliosides. It is important to note that the interpretation of the lipid composition of lipid mixtures containing glycosphingolipids (GSL) must be performed with great caution if 9-AA is used as a matrix. The reason is presumably the missing charge of (neutral) GSL [67], which renders them not detectable at standard MALDI conditions.

The detection of bacterial contaminations in food and biological samples is a very important issue of clinical chemistry. It is helpful that some glycolipids (such as endotoxins, also known as lipid A) are specific to dedicated bacteria and can be sensitively monitored by routine MALDI-MS [68]. Since the majority of these glycolipids are acidic compounds, alkaline matrices such as 9-AA are recommended [69]. Although the matrix norharmane is not commonly applied in routine lipid analysis [29], it is typically used to identify glycolipids in endotoxins and similar compounds [70]. It enhances the detectability of lipids such as monophosphoryl lipid A ten times in comparison to DHB and 100-fold compared to 9-AA. This is astonishing because DHB is normally much less sensitive than 9-AA [71]. Very recently, a method based on ZrO_2_ binding was developed, which enabled the separation between glyco- and phospholipids [72]. This is an important step toward the detection of low-abundant glycolipids and the prevention of the suppression of these lipids by more abundant lipids.

### 3.6. Problems Related to Mixture Analysis

The phenomenon of ion suppression is well known in soft-ionization MS. Although this may be beneficial because the intensities of unwanted matrix peaks can be reduced, this is normally a considerable problem if complex biological mixtures are analyzed: readily detectable compounds lead to the suppression of less sensitively detectable lipids. Typical examples are the suppression of PE by PC in the positive and the suppression of PE by sulfatides in the negative ion mode, respectively. This is not a problem of MALDI only, but ion suppression occurs with all soft-ionization MS techniques. Careful comparisons of different matrices, which are able to minimize this problem, are generally lacking, even if there are some recent studies in the context of MALDI-2 laser post-ionization experiments [73].

In particular, it was recently shown that CHCA could be converted into more effective matrix compounds [74]: cyano-naphthyl acrylic acid (CNAA) was particularly useful for phospholipid analysis in general, while cyano nitrophenyl dienoic acid (CNDA) favored the positive ion detection of neutral lipids such as DAG and TAG. Acidic PLs such as PI, PS, PA, and CL (and with slight reservations also PE) are detected with poor sensitivities as positive ions but are readily detected as negative ions. Although it would be of some interest, detection limits of the individual lipid classes are not often given in MALDI-based studies. This has different reasons:Although lipids are typically enriched by extraction with organic solvents, some salt is still present after extraction in the organic phase. The presence of these salts affects the detection of particularly acidic lipids [75].The age of the used mass spectrometer and potential contaminations (in both the ion source and the detector) negatively affect the detectability of the analytes. To the best of our knowledge, this particularly applies to negatively charged lipids. This renders the comparison of the data by different groups challenging.

Although this aspect is beyond the topic of this paper, standardization is an important issue in all “omics” approaches [76]. This particularly applies if no internal standards are used.

It was recently demonstrated that black phosphorus nanomaterial, in combination with a microchannel technique, effectively detects a variety of lipids. This approach detects particularly PC and lysophosphatidylcholines (LPC) with significant sensitivities: detection limits of about 0.2 μg/mL could be established. The method was applied for the detection of the serum PC/LPC ratios in child patients with asthma and other respiratory diseases. This MALDI approach has the particular advantage that the residual salt content of the sample had only a negligible impact [77].

The detection of PE in the presence of comparable amounts of PC is the most challenging aspect of MALDI lipid analysis and aggravates the analysis of biological samples. PE is particularly suppressed in the positive (vide supra) ion mode, but even the peak assignments (based on the observed *m*/*z* ratios) may also be difficult if an inappropriate matrix is used. The most serious problem is the in-source fragmentation of one methyl group from the generated PC ions [78]. This methyl group loss renders the original PC ions detectable as negative ions, which can be mistaken as isomeric PE ions. These detection and/or assignment problems can be overcome in the following two ways [79].

#### 3.6.1. Separation of the Mixture into the Individual Lipid Classes

Chromatographic separation is an established tool in the analysis of complex mixtures, which minimizes the ion suppression effect. Virtually all lipids (which are not seen in the mixture) can be detected in the isolated fractions. Both HPLC and thin-layer chromatography (TLC), which is still widely used in lipid research [79], are valuable tools in the field. Nowadays, there are commercially available techniques that enable the direct combination between TLC and ESI MS by re-elution of the analytes from the TLC plate. Nevertheless, TLC can also be coupled to MALDI MS [80] at which the TLC plate is analyzed in a similar way as a thin tissue slice in MSI. Therefore, the TLC plate (or just the regions of interest) is dipped into the matrix solution and, after drying, “scanned” by MALDI MS. Although HPLC-MS is more widely used, there are increasing applications of TLC in order to analyze lipids [81]. A comprehensive review of this field is already available [80].

#### 3.6.2. Choosing the Most Suitable Matrix

A review dealing with useful matrices for MSI purposes has been recently published by the Hopf laboratory [12]. The data therein can be transferred to the acquisition of MALDI mass spectra of lipid solutions. As already mentioned above, PE is the most challenging lipid class: it is suppressed in the presence of other, more acidic lipids (such as PI or sulfatides) in the negative ion mode and particularly by PC in the positive ion mode. Due to a possible loss of a methyl group from the quaternary ammonium group, PC ions could be incorrectly assigned as isomeric PE species if no MS/MS experiments are performed [82]. Using the most appropriate matrix is very important to detect all lipid classes of interest. For instance, it is well known that TAG is only detected in the presence of DHB, but not 9-AA [71].

In addition to typical organic matrices (such as DHB), there are some peculiar matrices for the analysis of small molecules that also includes lipids with molecular weights smaller than 1000 g/mol [13]: ionic liquid matrices (ILM), i.e., salts between amines and CHCA are useful alternatives to classical organic matrices [83]. ILM typically lead to smaller spot sizes, which minimizes shot-to-shot variations and simultaneously decreases the extent of analyte/matrix cluster formation. Additionally, there is reduced fragmentation because the stability of lipids is enhanced in the presence of ILM. Nevertheless, ILM are so far only scarcely used, and further work is required to elucidate its applicability of ILM. A more detailed survey of these aspects is available in [83].

## 4. Summary

MALDI-TOF MS was originally designed to analyze polar macromolecules, particularly proteins, and peptides obtained by enzymatic digestion. However, due to the increased use of MSI studies, it became evident that lipids can also be conveniently studied by MALDI-MS. There are two important advantages of lipids: (i) they generate significant ions yield, i.e., they are detectable with reasonable sensitivities and (ii) lipids occur in all tissues in larger amounts. This has fostered the interest in lipid analysis by MALDI. Two timely papers which summarize the role of MALDI-MS in veterinary research [84] and in clinical analysis [85] were recently published.

Besides the instrument hardware, such as the type of the laser (emission wavelength) and the mass analyzer (often TOF), instrument settings (the laser fluence and the detector voltage), the matrix (often in combination with the solvent) has a tremendous effect on the quality of the achieved spectra. Users of MALDI-MS must be aware that selected lipid classes (such as TAG) are not detectable if an inappropriate matrix is used [4]. This may also be considered an advantage because some impurities can be suppressed at optimum conditions. Although there were [86] (and still are [74]) many studies to rationally design superior matrix compounds, many of these matrices did not fulfill the expectations. Thus, it has to be admitted that the finding of powerful MALDI matrices is normally an empirical task. Many of the most famous matrices, for instance, the widely applicable DHB matrix, were actually found by accident [87] and not through a sophisticated strategy. The vacuum stability (to avoid changes in the matrix/analyte ratio during the acquisition of the spectra), the absorbance at the laser wavelength (in order to enable gentle ionization), and the gas phase basicity/acidity can be used as coarse guidelines to estimate the quality of a chemical compound as MALDI matrix. There are some more sophisticated studies on the rational improvement of common cinnamic acid and its conversion into phenyl cinnamic acid derivatives [88]. Another example is the invention of 4-chloro-alpha-cyanocinnamic acid, which is a very sensitive matrix for protein analysis and enables the detection of chloramines by MALDI-MS [89]. In the case of free acids, an enhanced molecular weight of the matrix may also be beneficial [90].

Anyway, it would be a considerable breakthrough if the optimum matrix for a particular problem could be established exclusively by theoretical studies. This would be of significant importance regarding MALDI MSI: even nowadays, two different matrices are needed to obtain high-quality positive and negative ion spectra. This is laborious and time-consuming. It would be major progress in the field if different matrices were not needed. This aim is only achievable if rational “matrix engineering” gets possible.

## Figures and Tables

**Figure 1 biomolecules-13-00546-f001:**
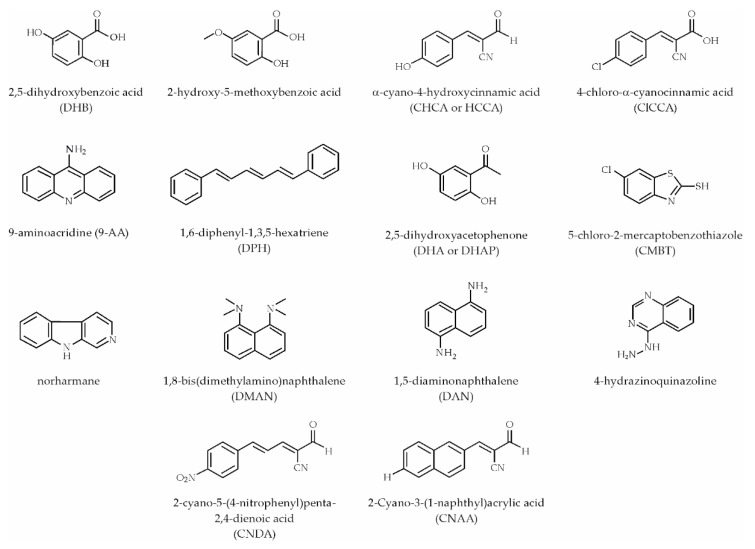
Survey of the organic MALDI matrices, which are discussed in this review. It is important to note that these are (in our opinion) just the most important matrices in the lipid field. For specific questions and certain analytes, the application of different matrices may be beneficial.

**Figure 2 biomolecules-13-00546-f002:**
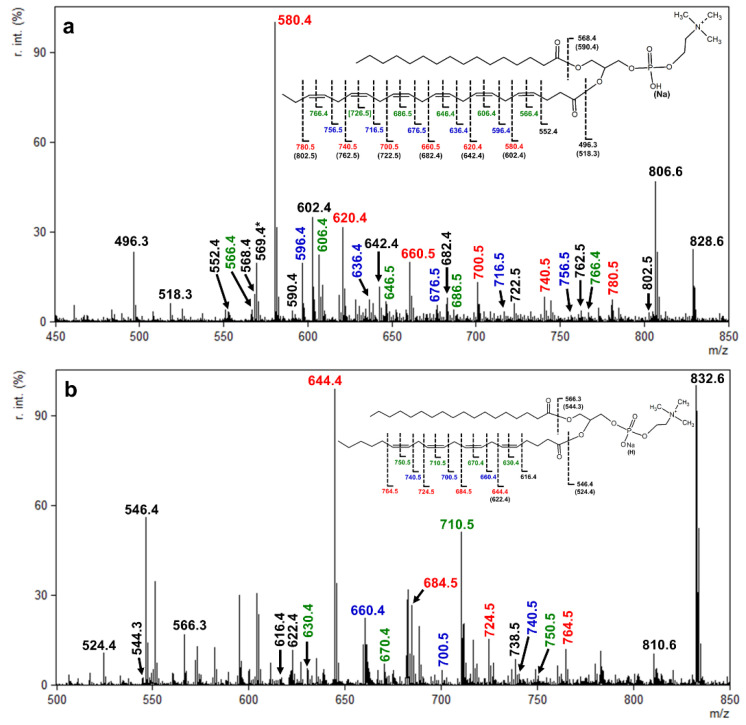
ISF-(in-source fragmentation)-MALDI/TOF spectra of PUFA-containing PCs. (**a**) ISF MALDI/TOF spectrum of PC 16:0/22:6. Protonated product ions from vinylic, double-bond, and allylic cleavages were marked in red, green, and blue, respectively, in the spectrum and the structure inset. Parenthesized *m*/*z* values in the structure inset denoted the sodiated form, and the bracketed *m*/*z* value denoted undetected species. (**b**) ISF MALDI/TOF spectrum of PC 18:0/20:4 with added Na^+^ in preparation. Sodiated product ions from vinylic, double-bond, and allylic cleavages were marked in red, green, and blue, respectively, both in the spectrum and the structure inset. The *m*/*z* values of protonated ions were parenthesized in the structure inset. Reproduced (with slight modifications) from [56] with permission from Springer and courtesy by the authors. The asterisk denotes an ions stemming from the used DHB matrix.

## Data Availability

Not applicable.

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
