# Peer review of "A Five-Year Update on Matrix Compounds for MALDI-MS Analysis of Lipids"

_biomolecules, 2023, doi:10.3390/biom13030546_

Round 1
Reviewer 1 Report
This is a good review on MALDI matrices using in lipid analysis. I have several suggestions and comments.
Line 83-85: This statement is not accurate. There is substantially more analysis of lipids than DNA by MALDI MS.
Line 163-168. As this is a review paper on MALDI matrices in lipid research, here would be a good point to state what MALDI matrices have been used for TAG analysis.
Line 253. Add which MALDI matrices have been used for PC oxidation product analysis
Line 398. Has anyone performed such engineering yet? I understand that choosing an optimal MALDI matrix is by try and error principle, as we still don’t fully understand MALDI ionization mechanisms.
Author Response
Reviewer #1
This is a good review on MALDI matrices using in lipid analysis. I have several suggestions and comments.
Thank you for the appreciation of our work and your kind comments. We have addressed all your suggestions and comments. We hope that you will be satisfied with the way we have dealt with your comments.
Line 83-85: This statement is not accurate. There is substantially more analysis of lipids than DNA by MALDI MS.
This is and we apologize for this misunderstanding and/or the imprecise statement. Of course, MALDI was (and still is) particularly used in the field of protein and peptide analysis. This is the "classical" application and still widely used. The interest in lipids has just recently grown - with the development of MALDI MS imaging. We tried our very best to clarify these aspects.
Line 163-168. As this is a review paper on MALDI matrices in lipid research, here would be a good point to state what MALDI matrices have been used for TAG analysis.
Thank you for this comment. Some particularly useful matrices for the analysis of triacylglycerols (TAG) are now explicitly mentioned. According to our best knowledge, 2,5-dihydroxybenzoic acid (DHB) is the classical most established matrix in this field.
Line 253. Add which MALDI matrices have been used for PC oxidation product analysis
Thank you again for the careful reading of our manuscript and the appropriate comment. We have added some comments and specified some useful matrix compounds. It is now also explicitly mentioned that derivatization is another useful approach to improve the detection of small quantities of PC oxidation products in biological lipid mixtures. This particularly applies for chain-shortened aldehydes, which can be rather easily derivatized.
Line 398. Has anyone performed such engineering yet? I understand that choosing an optimal MALDI matrix is by try and error principle, as we still don’t fully understand MALDI ionization mechanisms.
It is true that there are so far not so many attempts to rationally predict the optimum matrix for a particular application. Commonly used parameters are the vacuum stability, the absorption at the laser wavelength, the ion affinity and many others. There are some studies on the improvement of classical cinamic acid and their intentional conversion into phenyl cinnamic acid derivatives (DOI: 10.1007/s00216-016-0096-6). Another example is the invention of 4-chloro-alpha-cyanocinnamic acid, which is a very sensitive matrix for protein analysis and enables the detection of chloramines by MALDI MS (DOI: 10.1073/pnas.0803056105). Another example is the optimization of the DMAN (1,8-bis(dimethyl-amino)naphthalene) matrix to increase its stability in the high vacuum and to improve its application for the analysis of free fatty acids (DOI: 10.1021/ac901048z). This information is now explicitly provided and the corresponding references given.
Reviewer 2 Report
The authors well summarizes and shows the results of research on MALDI-TOF MS on lipids analysis over the past five years.
Recently, many studies have been conducted to obtain energy and fuel from algae. A typical bioenergy material obtained from algae is the lipids family including polyunsaturated fat acid. Therefore, it is recommended that the author further summarizes and presents the results of recent studies that analyzed PUFA and Lipids with MALDI-TOF MS from a bioenergy perspective (e.g., ACS Omega 2022, 7, 24785-24794 etc.).
1. What is the main question addressed by the research? --> Authors well discribe that MALDI matrices for analyzing of lipid materials. 2. Do you consider the topic original or relevant in the field? Does it address a specific gap in the field? --> This review of the matrix described by the author well illustrates recent efforts in the field to efficiently analyze lipids. 3. What does it add to the subject area compared with other published material? --> It has already been presented in the previous my review comment. 4. What specific improvements should the authors consider regarding the methodology? What further controls should be considered? --> There is no need for additional improvements in methodology, except the previous my review comment. However, the abstract in manuscript is poor. Reinforce abstract by summarizing the text well. 5. Are the conclusions consistent with the evidence and arguments presented and do they address the main question posed? --> Yes 6. Are the references appropriate? --> Yes 7. Please include any additional comments on the tables and figures. ** --> There is no additional comments on the tables and figures
Author Response
The authors well summarizes and shows the results of research on MALDI-TOF MS on lipids analysis over the past five years.
Thank you for your kind appreciation of our manuscript and the careful reading. Thank you as well for the helpful and kind comments. We have tried our very best to deal carefully with your comments and were - in our opinion - able to address all your concerns.
Recently, many studies have been conducted to obtain energy and fuel from algae. A typical bioenergy material obtained from algae is the lipids family including polyunsaturated fat acid. Therefore, it is recommended that the author further summarizes and presents the results of recent studies that analyzed PUFA and Lipids with MALDI-TOF MS from a bioenergy perspective (e.g., ACS Omega 2022, 7, 24785-24794 etc.).
This is a reasonable suggestion and we agree that algal lipids are of increasing importance - as potential fuel and for (human) nutrition due to the high content of unsaturated fatty acids. Therefore, this aspect is now discussed in more detail in our revised manuscript and some additional references are given. The reference indicated by the reviewer is now also mentioned - although the performed investigations are not based on MALDI MS but on ESI MS. MALDI MS is only mentioned a single time in this paper and just a single reference is given.
- What is the main question addressed by the research?
Authors well discribe that MALDI matrices for analyzing of lipid materials.
Thank you for the kind statement.
- Do you consider the topic original or relevant in the field? Does it address a specific gap in the field?
This review of the matrix described by the author well illustrates recent efforts in the field to efficiently analyze lipids.
Thank you for the appreciation of our work.
- What does it add to the subject area compared with other published material?
It has already been presented in the previous my review comment.
We have made no changes since no changes were suggested.
- What specific improvements should the authors consider regarding the methodology? What further controls should be considered?
There is no need for additional improvements in methodology, except the previous my review comment. However, the abstract in manuscript is poor. Reinforce abstract by summarizing the text well.
Thank you again for careful reading of our manuscript. In the light of your comment, we do now also see some weaknesses of the previous version of the abstract. The abstract was completely rewritten and we hope that you will be satisfied with the new version.